# Evolving a Model for Cochlear Implant Outcome

**DOI:** 10.3390/jcm12196215

**Published:** 2023-09-26

**Authors:** Ulrich Hoppe, Anne Hast, Joachim Hornung, Thomas Hocke

**Affiliations:** 1Cochlear Implant Center CICERO, Department of Otorhinolaryngology-Head and Neck Surgery, Uniklinikum Erlangen, Waldstr. 1, D-91054 Erlangen, Germany; anne.hast@uk-erlangen.de (A.H.); joachim.hornung@uk-erlangen.de (J.H.); 2Cochlear Deutschland GmbH & Co. KG, Mailänder Str. 4a, D-30539 Hannover, Germany; thocke@cochlear.com

**Keywords:** word recognition, CI outcome, prediction, generalised linear model, adults

## Abstract

Background: Cochlear implantation is an efficient treatment for postlingually deafened adults who do not benefit sufficiently from acoustic amplification. Implantation is indicated when it can be foreseen that speech recognition with a cochlear implant (CI) is superior to that with a hearing aid. Especially for subjects with residual speech recognition, it is desirable to predict CI outcome on the basis of preoperative audiological tests. Purpose: The purpose of the study was to extend and refine a previously developed model for CI outcome prediction for subjects with preoperative word recognition to include subjects with no residual hearing by incorporating additional results of routine examinations. Results: By introducing the duration of unaided hearing loss (DuHL), the median absolute error (MAE) of the prediction was reduced. While for subjects with preoperative speech recognition, the model modification did not change the MAE, for subjects with no residual speech recognition before surgery, the MAE decreased from 23.7% with the previous model to 17.2% with the extended model. Conclusions: Prediction of word recognition with CI is possible within clinically relevant limits. Outcome prediction is particularly important for preoperative counseling and in CI aftercare to support systematic monitoring of CI fitting.

## 1. Introduction

Cochlear implantation is an efficient treatment for postlingually deafened adults with severe and profound hearing loss. In particular, a cochlear implant (CI) is indicated when the benefit from acoustic amplification is insufficient [1,2,3,4,5,6,7]. For mild and moderate hearing loss, a hearing aid (HA) is the option of choice, while for higher degrees of hearing loss, it must be carefully considered which approach is better. Especially in the transition range, i.e., hearing thresholds better than 80 dB_HL_ (dB hearing loss), the variability of the aided speech recognition is substantial [8,9,10,11,12,13,14,15,16,17]. Nevertheless, in individual cases the speech recognition with HA can be assessed preoperatively. However, the large variability in CI outcome as assessed by word recognition scores with CI [18,19,20,21,22] represents a major obstacle: for the patient population with benefit from HAs, the *individual* prediction is of major importance, as the patient and the professional have to balance the residual aided word recognition with the HA, the expected word recognition with CI, the expected improvement in quality of life, and the impact of CI surgery. Some studies have also included subjects with lesser hearing loss (e.g., <80 dB_HL_) who were considered likely to benefit from cochlear implantation [5,6,17,23,24,25,26,27,28,29]. A retrospective analysis [22] of 312 postlingually deafened adult CI recipients yielded the preoperative maximum word recognition score (WRS_max_) as a predictor for the minimum WRS with CI at conversation level, WRS_65_(CI). The importance of this preoperative measure was confirmed by two studies including, respectively, 128 [28] and 664 [17] cases. In an earlier study we addressed explicitly the prediction of WRS_65_(CI) in a population with hearing losses of less than 80 dB_HL_ only [6]. This retrospective analysis led to a generalised linear model (GLM) that provides an estimated prediction of WRS_65_(CI) six months after implantation on the basis of three preoperatively known factors: WRS_max_, the patient’s age at implantation, and the aided WRS at conversation level, WRS_65_(HA), according to Equation (1).
(1)$${WRS}_{65}\left(CI\right)\left[\%\right]=\frac{100}{1+e^{-\left(\beta_{0}+\beta_{1}\cdot {WRS}_{max}+\beta_{2}\cdot Age+\beta_{3}\cdot {WRS}_{65}\left(HA\right)\right)}}$$
with β_0_ = 0.84 ± 0.18, β_1_ = 0.012 ± 0.0015, β_2_ = −0.0094 ± 0.0025 year^−1^, and β_3_ = 0.0059 ± 0.0026; all WRS expressed in %.

Figure 1 illustrates the characteristics of this GLM. WRS_max_ accounts for up to 27 percentage points (pp) in WRS_65_(CI) differences. WRS_65_(HA) influences the prediction by up to 9 pp, while age at implantation is associated with a deterioration of up to 17 pp. The GLM resulted from an analysis based on a population of 128 postlingually deafened adult CI recipients, all with a preoperative hearing loss equal to or less than 80 dB_HL_ as measured by the hearing loss at 0.5, 1, 2, and 4 kHz (four-frequency pure-tone average, 4FPTA).

The prediction error of the model as described by the median absolute error (MAE) was found to be 13.5 pp [6], with one-quarter of the study population scoring 12 pp or more below prediction. A subsequent prospective study [29] confirmed the applicability of the model for CI recipients within certain boundary conditions: for a patient population with a preoperative WRS_max_ greater than zero, a prediction error of 11.5 pp was found. Only 6% (5/85) of the recipients missed the predicted score by more than 20 pp within one year after implantation. As shown in Figure 1, the output range is limited to scores between 49% and 90%. This is due to the fact that patients with significant residual hearing are most likely to perform in this range [6,17,22,28,29]. This is not the case for the application of the model in a population with preoperative WRS_max_ = 0%, which, as expected, resulted in a higher prediction error of 23.2 pp. If both WRS_max_ and WRS_65_(HA) are zero, the prediction from Equation (1) is based solely on the patient’s age, represented by β_2_, and the population mean outcome, represented by β_0_.

While in some previous analyses duration of deafness (DoD) played a significant role [19], DoD was not included in the model (Equation (1)). This is due to the fact that only subjects with hearing threshold better than 80 dB_HL_ were included in the previous study [6]. Holden et al. [20] showed that the duration of hearing impairment (DHI) is a factor that contributes to speech recognition with CI. Additionally, DHI is applicable for subjects with residual hearing, regardless of the degree of hearing loss.

The goal of this study was the extension and evolution of the model [6] in order to improve prediction, especially for patient populations with a preoperative WRS_max_ of zero and all degrees of hearing loss. The design requirements for the model were defined as follows: Since Equation (1) has proved its applicability [29,30], the coefficients β_0–3_ remained fixed. Only preoperative measures were to be included in the model. Additionally, these measures were to be subsets of clinical routine measurements within the CI candidate assessment according to the German CI Guidelines [3] and the German white book CI provision [4].

## 2. Materials and Methods

### 2.1. Patients

In this study we evaluated data from all postlingually deafened adult patients who were provided with a Nucleus CI (Cochlear Ltd., Sydney, Australia) in the period April 2020 to December 2022 at the Ear, Nose, and Throat Clinic within the department of Head and Neck Surgery at the University Hospital of Erlangen. All study participants were native German speakers. The CI indication was in accordance with the current German CI guidelines [3]. All participants suffered from sensorineural or mixed hearing loss in the ear to receive the implant. They took part in our rehabilitation programme for a minimum of six months. Rehabilitation was performed in accordance with the most recent recommendations [3,31,32]. Rehabilitation includes speech processor fittings, auditory training, psychological counselling, and medical checks on a regular basis. Usually, rehabilitation is performed for more than twelve months.

Exclusion criterion was cognitive impairment that would have influenced the performance of the speech audiometry. Reimplantations were excluded. Postoperative WRS_65_(CI) for a period of at least six months after surgery and CI fitting were available for 165 patients. The patient population consisted of 90 men and 75 women. Their mean age at the time of surgery was 66 ± 14 years. The hearing loss for air conduction was determined as the mean value over the four octave frequencies 0.5, 1, 2, and 4 kHz (4FPTA). For hearing thresholds beyond the maximum possible presentation levels of the audiometers, a value of 130 dB_HL_ was imputed. The resulting mean preoperative hearing loss was 94 ± 21 dB_HL_. The 165 CI recipients used either the behind-the-ear processor CP1000 (or later) or the off-the-ear processor CP950 (or later). CI-aided listeners were divided into two groups according to their preoperative WRS_max_. Group 1 (*n* = 109) comprised individuals with WRS_max_ > 0%, while group 2 (*n* = 56) comprised those with WRS_max_ = 0%. While there were no significant differences between these groups in age or in duration of hearing impairment, audiometric data differed owing to the group definitions. Demographic details are summarised in Table 1. Figure 2 complements the characteristics in Table 1 by representing the individual data for age, duration of hearing loss, and duration of unaided hearing loss. Age was not correlated with either duration of hearing impairment (DHI) or duration of unaided hearing impairment (DuHI), while DHI and DuHI were strongly correlated (R_Spearman_ = 0.68 with *p* = 5 × 10^−24^).

### 2.2. Speech Audiometry

Speech recognition was assessed by the Freiburg monosyllable and Freiburg two-digit-number tests. The monosyllable test comprises 20 groups of 20 monosyllabic German nouns each; the number test comprises multisyllabic two-digit numbers in 10 groups of 10 numbers each (e.g., 98 was read as “achtundneunzig”) [33,34]. Usually, the numbers are understood much better than the monosyllabic words. Recognition rates correspond to low-frequency hearing thresholds [35]. The monosyllable test was used to determine the maximum word recognition score (WRS_max_), i.e., the word recognition score at the greatest just-tolerable sound pressure level or, in case of 100%, at lower levels. Additionally, WRS_65_(HA) was defined as the word recognition score with hearing aid measured at 65 dB_SPL_. The hearing aids were checked technically in advance. In particular, in situ measurements were performed to ensure that the settings yielded the necessary gains [16].

For the Freiburg two-digit numbers, the sound levels were adjusted individually in the range from 30 to 120 dB_SPL_ in 5 dB steps in order to find the sound pressure level for 50% recognition (SRT_num_). For SRTs above the maximum possible presentation levels of the audiometers, a value of 120 dB_HL_ was imputed. All audiometric measurements were performed monaurally with the ear that was intended for the implant, while the contralateral ear was masked appropriately when necessary.

The 4FPTA was calculated from the pure-tone audiometry data as the mean value of the hearing threshold at 0.5, 1, 2, and 4 kHz.

For the postoperative measurements, the word recognition score with CI system in free field at 65 dB_SPL_, WRS_65_(CI) was assessed. The free-field measurements were conducted in a soundproof cabin measuring 6 × 6 m. The loudspeaker was placed 1.5 m in front of the patient (0° azimuth). The contralateral ear was masked appropriately with broadband noise introduced through headphones, if necessary.

### 2.3. Data Analysis

The software Matlab (MathWorks, Natick, MA, USA) version R2019b was used for all calculations and figures. A GLM was applied to the data to predict WRS_65_(CI); this model represented a further development of our earlier model (see Section 1) and is described below. Significant differences in word recognition scores were determined according to the characteristics of the Freiburg monosyllable test [36].

## 3. Results

### 3.1. Preoperative Measurements

Figure 3A–C show the interrelationships between word recognition scores, WRS_65_(HA) and WRS_max_, and the average pure-tone hearing loss, 4FPTA. Figure 3D–F show how speech recognition thresholds for numbers in quiet (SRT_num_) are related to the 4FPTA and the two WRS. The curves in Figure 3A,B represent WRS as a function of 4FPTA in a population of HA users from previous studies [8,14]. In all cases the preoperative WRS_65_(HA) was within the current German CI guidelines [3], which recommend a cut-off at 60% for preoperative WRS_65_(HA). Figure 3D–F illustrate the relationship between the audiometric measures WRS_65_(HA), WRS_max_, 4FPTA, and SRT_num_. Even for cases where WRS_65_(HA) and WRS_max_ are zero, SRT_num_ can still be measured: there were 95 of 165 cases with WRS_65_(HA) = 0%, of which 53 (56%) had a measurable SRT_num_. Among the 56 cases in group 2 (preoperative WRS_max_ = 0%), a measurable SRT_num_ was still found in 12 cases (21%). All speech recognition measures were highly correlated: WRS_max_ with SRT_num_ (R_Spearman_ = −0.72 with *p* = 6 × 10^−28^), WRS_65_(HA) with SRT_num_ (R_Spearman_ = −0.58 with *p* = 2 × 10^−16^), and WRS_max_ with WRS_65_(HA) (R_Spearman_ = −0.62 with *p* = 1 × 10^−18^).

### 3.2. Postoperative Measurements

Figure 4 illustrates the relationship between the two preoperative WRS and WRS_65_(CI) six months after surgery. The two groups with WRS_max_ above zero (group 1, black) or equal to zero (group 2, blue) show WRS_65_(CI) ranging from 0 to 100%. Both groups have their peak in Figure 4C at a WRS_65_(CI) of 70%, and the median for WRS_65_(CI) is 70% for both groups. However, the variabilities differed considerably: the standard deviation of WRS_65_(CI) was 19 pp for group 1 and 30 pp for group 2. Postoperative results are summarised in Table 2.

With respect to the minimum predictor [22] for WRS_65_(CI), this study yielded the following results for group 1: Six months after surgery, 6 cases (5.5%) did not reach WRS_max_, while in 64 cases (58.7%) WRS_max_ was significantly [36] exceeded. The remaining 39 (35.8%) cases reached WRS_max_ within the confidence intervals yielded by the Freiburg test [36].

### 3.3. Model Expansion

GLMs were applied to the complete data set. As additional input variables to the previous model [6] (see Equation (1)), the duration of hearing impairment (DHI), the duration of unaided hearing impairment (DuHI (a subperiod of DHI)), and SRT_num_ were considered. The strong correlations between SRT_num_ and WRS_max_ and between SRT_num_ and WRS_65_(HA) (see Section 3.1) indicate that the linear equation system is over-determined.

All regressions with SRT_num_ included resulted in a GLM with a corresponding positive β_i_. Such an equation would result in a poorer prediction for WRS_65_(CI) with better preoperative SRT_num_. The ablation analysis [37] did not yield an improvement with respect to the overall MAE with SRT_num_ included (12 pp) compared with the final GLM (12 pp). Consequently, the regression analysis was continued with DHI and DuHI only. The ablation analysis yielded the best results applying the two predictors with an interaction term [38]: The overall MAE was 12.3 pp, while for group 1 it was 11.1 pp and for group 2 it was 17.0 pp. Table 3 summarises the results of the regression analysis.

On the basis of Table 3 and with removal of the two non-significant contributors DHI and the interaction term DHI:DuHI, Equation (1) expands to
(2)$${WRS}_{65}\left(CI\right)\left[\%\right]=\frac{100}{1+e^{-\left(\beta_{0}+\beta_{1}\cdot {WRS}_{max}+\beta_{2}\cdot Age+\beta_{3}\cdot {WRS}_{65}\left(HA\right)+\beta_{0}^{\prime }+\beta_{4}\cdot DuHI\right)}}$$
with $\beta_{0}$ = 0.84 ± 0.18, $\beta_{1}$ = 0.012 ± 0.0015, $\beta_{2}$= −0.0094 ± 0.0025 year^−1^, $\beta_{3}$ = 0.0059 ± 0.0026, $\beta_{0}^{\prime }$ = 0.35 ± 0.04, and $\beta_{4}$ = −0.0171 ± 0.0056 year^−1^ (all WRS expressed in %), as shown in Table 3.

The application of Equation (2) to the study population yields prediction errors as shown in Figure 5, separately for the two groups. A comparison with the previous model [6] is displayed as well.

In cases where DuHI is not available in the clinical data set, a regression analysis with DHI would be helpful. Consequently, this was done. Table 4 summarises the results of this analysis. On the basis of Table 4, Equation (1) expands to
(3)$${WRS}_{65}\left(CI\right)\left[\%\right]=\frac{100}{1+e^{-\left(\beta_{0}+\beta_{1}\cdot {WRS}_{max}+\beta_{2}\cdot Age+\beta_{3}\cdot {WRS}_{65}\left(HA\right)+\beta_{0}^{\prime }+\beta_{4}\cdot DHI\right)}}$$
with $\beta_{0}$ = 0.84 ± 0.18, $\beta_{1}$ = 0.012 ± 0.0015, $\beta_{2}$= −0.0094 ± 0.0025 year^−1^, $\beta_{3}$ = 0.0059 ± 0.0026, $\beta_{0}^{\prime }$ = 0.41 ± 0.04, and $\beta_{4}$ = −0.0125 ± 0.0013 year^−1^ (all WRSs expressed in %), as shown in Table 4.

The resulting MAEs are 11.3 pp for group 1 and 18.8 pp for group 2, with an overall MAE of 14.0 pp.

## 4. Discussion

The vast majority (89%) of the patients included in this study showed signifearimproved speech recognition without any patient experiencing a lower WRS six months after cochlear implantation.

Both populations (patients with preoperative WRS_max_ larger than zero, group 1, and patients with preoperative WRS_max_ equal to zero, group 2) showed a median WRS_65_(CI) of 70%. However, as illustrated by Figure 4, the variability of the outcome was greater for group 2, and the mean WRS_65_(CI) was smaller: 59% in group 2, compared with 68% in group 1. Additionally, group 2 includes seven subjects (13%) with WRS_65_(CI) = 0, while in group 1, only one subject (1%) scored 0%. These subjects clearly indicate the demand for future studies dealing with unexpected low speech perception.

The extension of the prediction model for CI outcome in CI recipients with preoperative WRS_max_ = 0 is feasible. It was shown that for group 2 an improved prediction is possible without impairment of the prediction for group 1. Most remarkably, the inclusion of just one additional input variable (the duration of unaided hearing impairment, DuHI) in the previous prediction model for the WRS_65_(CI) [6] resulted in a decreased prediction error for group 2: the new GLM (Equation (2)) resulted in a decreased MAE of 17.0, compared with the MAE of the previous model (Equation (1)) of 23.7 pp. The prediction error for group 1 remained almost unchanged: the new model indicates a slightly decreased MAE of 11.1 pp, compared with 11.4 obtained from the previous model [6].

The results demonstrate that it is possible to use one model for both groups. This enables a seamless application for all CI candidates independently from the preoperative speech recognition. It may be used as a baseline for further refinements of the model for specific candidate groups. However, this would require by far a greater number of cases. The durations of hearing impairment and unaided hearing impairment, DHI and DuHI, were found to be strongly correlated (R_Spearman_ = 0.7). Hence, they may provide similar information on the CI outcome. The ablation analysis showed that the MAE was not greatly increased when DHI or DuHI was omitted. We decided to retain the latter because DHI was found as not significant (*p* = 0.16) in the presence of DuHI. Additionally, the MAE was smaller for both groups when DuHI was used instead of DHI (Equation (2)). However, the DHI offers some advantages. The DHI is just defined by one time point, the time of onset of hearing loss, while determination of DuHI requires knowledge of two time points: HA provision and onset of hearing loss. Yet both factors depend on the patient’s ability to remember or reconstruct events which may well have occurred decades earlier. In summary, the model according to Equation (3) inherits larger MAE. However, Equation (3) and therefore the DHI may be used in cases where DuHI is not available.

In this study, the DHI replaces the previously used duration of deafness, DoD. Though DoD was frequently used in CI studies, it is not well defined; regarding DoD, an obsolete classification [39] refers to a cut-off of 81 dB_HL_ for the grade “profound impairment including deafness”. A more recent classification from WHO [40] defines “Complete or total hearing loss/deafness” as hearing threshold in the better ear of 95 dB_HL_ or greater. Those authors explicitly explain that the PTA should not be used as the “sole determinant for rehabilitation” and that “the classification and grades are for epidemiological use” [40]. For prediction models and clinical process management, to our knowledge, there is a lack of applicable, defined criteria for cut-off relating to the duration of deafness and hearing impairment. Additionally, in the presence of a decentralised hearing health care system (e.g., in Germany), the chance of obtaining all necessary data retrospectively is rather low. In our population of consecutive Nucleus CI provisions in adults within a period of 2.5 years, the majority is not deaf using this definition, so a broader application of DoD in a regression model is not relevant. In addition, about one-third of the patients defined as “deaf” using the above WHO criterion [40] had a measurable ipsilateral maximum recognition score for Freiburg words, and slightly under one-half had a measurable speech recognition threshold for Freiburg numbers in quiet. This supports the preference for functional, speech-related variables instead of DoD.

There was a slight decrease in MAE for group 1 only (preoperative WRS_max_ > 0). This can be interpreted as giving strong support to the use of WRS_max_ for predictive purposes [6,17,22,28], as it accumulates the detrimental effects of long DHI (or DuHI). The situation is different in group 2 (preoperative WRS_max_ = 0), where such functional assessment with the established test WRS_max_ and WRS_65_(HA) is not possible. Here, the additional information of DuHI or DHI considerably reduces the prediction error.

According to the design requirements, the regression analysis using the GLM was conducted across all data by using all data in a first attempt. The effects of these three variables upon the prediction error are different. It was found that SRT_num_ did not decrease MAE. Hence, SRT_num_ was not taken into account any longer, which however does not necessarily mean that this variable is unimportant. Together with the strong correlation with WRS_max_ and WRS_65_(HA), this indicates an over-determined equation system. Nevertheless, especially for cases with no preoperative monosyllable speech perception, it might be a useful addition. In our population only about one-quarter of group 2 had a measurable SRT_num_. Perhaps an additional split beyond groups 1 and 2 will improve the prediction with the help of SRT_num_ in a clearly and more narrowly defined population. On the other hand, other model approaches—such as random forest regression—would induce such a split per se. However, more data would be needed for such an approach. In a recent study, Rieck et al. [17] used the Freiburg numbers and found a predictive value in a population of nearly 500 recipients. Two characteristics of their study population would support the assertion of a positive impact of SRT_num_ on prediction error in a population with low preoperative speech perception in general. The mean values obtained in their study represent the characteristics of an established patient population with a preoperative mean WRS_65_(HA) of 4.2% compared with 9.7% and a WRS_max_ of 11.8 compared with 27.6% in this population. Rieck et al. [17] included clinical data with implantations dating from 2002 to 2019, while the inclusion period of the present study was from 2020 to 2022. Consequently, this relationship should be reconsidered in future studies that include more CI candidates who are in group 2 but who have measurable SRT_num_.

## 5. Conclusions

Cochlear implantation can be considered if speech recognition with hearing aids is insufficient. This applies also for patients with pure-tone hearing loss in the range of 60 dB_HL_. The preoperative prediction of expected word recognition after CI provision is possible within clinically relevant limits.

Less variable results for postoperative word recognition were observed in patients with preoperative maximum word recognition greater than zero (group 1) compared with patients without preoperative maximum word recognition (group 2).

The inclusion of additional model input variables—‘duration of hearing impairment’ or ‘duration of unaided hearing impairment’—to the variables ‘word recognition scores‘ and ‘age at implantation‘ already used in the model resulted in decreased prediction errors for group 2. However, the prediction error in group 2 was still larger than in group 1. In group 1 the inclusion of additional input variables did not result in a lower prediction error.

We believe that this model will be applicable in preoperative counseling (with a higher accuracy in group 1 than in group 2) and will also be useful in CI aftercare to support the systematic monitoring of CI fitting that is conducted to optimise postoperative adjustment.

## Figures and Tables

**Figure 1 jcm-12-06215-f001:**
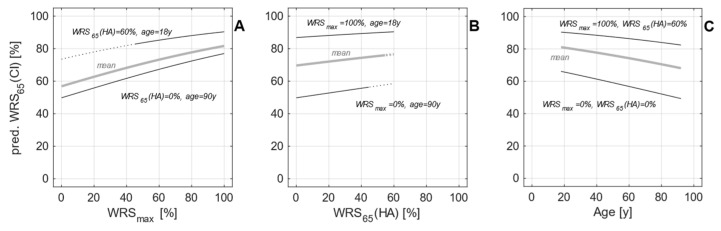
Output characteristics of the generalised linear model as a function of the three input variables within a reasonable input range. The predicted word recognition score WRS_65_(CI) after six months is shown as a function of (**A**) preoperative maximum word score WRS_max_, (**B**) preoperative aided score WRS_65_(HA), and (**C**) age at implantation. In each panel the remaining two factors are kept constant at the selected values indicated, covering the observed range, and the thin black curves show the variation in WRS_65_(CI). The thick grey curves represent the model’s results for the most recent population means at our clinic: WRS_max_ = 50%, WRS_65_(HA) = 9%, and age = 66 years. Dotted lines indicate a rather unlikely combination of input factors, namely a high WRS_65_(HA) in the presence of much lower WRS_max_.

**Figure 2 jcm-12-06215-f002:**
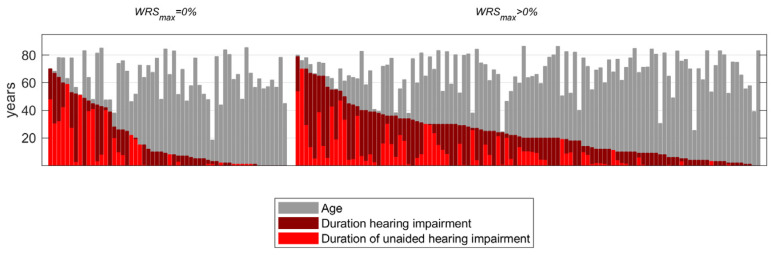
Distribution of age, duration of hearing impairment, and duration of unaided hearing impairment in the two patient groups with preoperative maximum word recognition (WRS_max_) of zero (**left**) or above zero (**right**).

**Figure 3 jcm-12-06215-f003:**
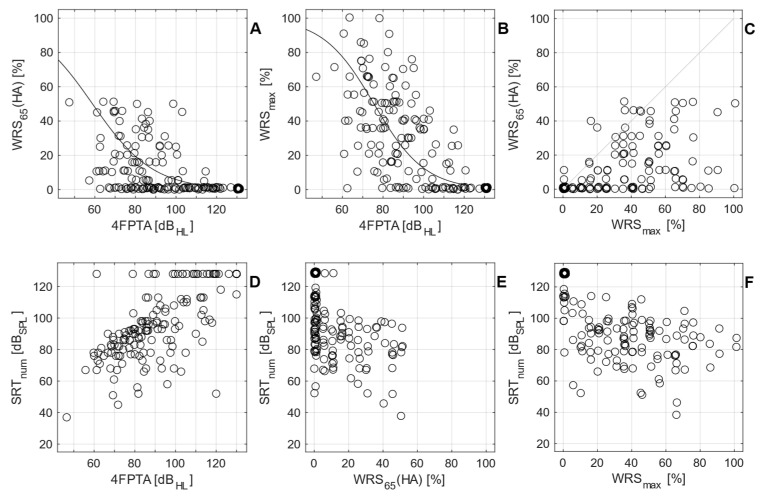
Preoperative audiometry of the 165 cases: (**A**) The aided word recognition score, WRS_65_(HA), as a function of average pure-tone hearing loss, 4FPTA; (**B**) the maximum word recognition score, WRS_max_, as a function of 4FPTA; (**C**) relation between WRS_65_(HA) and WRS_max_. The black curves in panels (**A**,**B**) represent the average relation between WRS values and 4FPTA in a population of HA users [8,14]. The lower panels (**D**–**F**) show the relationship between SRT_num_ and 4FPTA, WRS_65_(HA), and WRS_max_, respectively.

**Figure 4 jcm-12-06215-f004:**
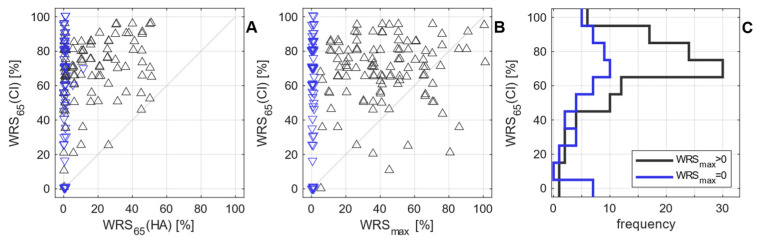
Relationship between preoperative and postoperative word recognition scores for the 165 cases: (**A**) word recognition score with CI after six months (WRS_65_(CI) vs. the preoperative aided score, WRS_65_(HA)); (**B**) WRS_65_(CI) vs. the maximum preoperative word recognition score, WRS_max_; (**C**) distribution of WRS_65_(CI) for the two patient groups (black, group 1 with a preoperative WRS_max_ > 0%; blue, group 2 with WRS_max_ = 0%). This colour code applies to panels (**A**,**B**) as well.

**Figure 5 jcm-12-06215-f005:**
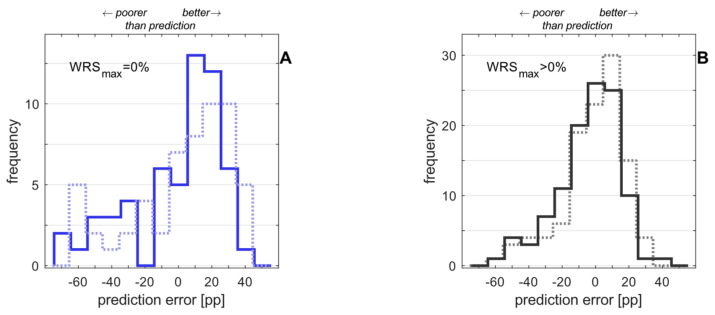
Distribution of differences between predicted and measured word recognition with CI and WRS_65_(CI), six months after surgery: (**A**) prediction errors in percentage points (pp) for group 1 (preoperative WRS_max_ > 0); (**B**) prediction errors for group 2 (WRS_max_ = 0). In both panels the dotted line indicates the prediction error resulting from the application of the previous model [6] (Equation (1)). The solid line indicates the prediction error resulting from the advanced model (Table 3). The MAE was 11.1 pp in group 1 and 17.1 pp in group 2 according to Equation (2). The previous model resulted in MAEs of 11.3 pp for group 1 and 23.7 pp for group 2.

**Table 1 jcm-12-06215-t001:** Patient characteristics.

	Size	Age [Years]	4FPTA [dB_HL_]	WRS_max_ [%]	WRS_65_(HA)	Duration of Hearing Impairment [Years]	Duration of Unaided Hearing Impairment [Years]	SRT_num_ [dB_SPL_]
Group 1 WRS_max_ > 0%	109	67 ± 14	83 ± 14	42 ± 23	15 ± 16	24 ± 18	9 ± 13	85 ± 15
Group 2 WRS_max_ = 0%	56	64 ± 14	114 ± 17	0	0 ± 1	20 ± 22	10 ± 16	124 ± 10
total	165	66 ± 14	94 ± 21	27 ± 27	10 ± 15	22 ± 20	9 ± 14	98 ± 23

SRT_num_, speech recognition threshold for 50% number recognition; SPL, sound pressure level; for other abbreviations, see text above. Means ± standard deviations are shown.

**Table 2 jcm-12-06215-t002:** Variability of word recognition with CI six months after surgery with respect to preoperative maximum word recognition.

Group	Size	WRS_65_(CI) [%]		No. of Cases with a Score of …		
		Mean ± SD	Median	WRS_65_(CI) = 0%	WRS_65_(CI): >0–<50%	WRS_65_(CI): 50–100%
Group 1 WRS_max_ > 0%	109	68 ± 19	70	1 (1%)	12 (11%)	96 (88%)
Group 2 WRS_max_ = 0%	56	59 ± 30	70	7 (13%)	9 (16%)	40 (71%)
total	165	65 ± 24	70	8 (5%)	21 (13%)	136 (82%)

**Table 3 jcm-12-06215-t003:** Results of regression analysis with two additional predicting variables and their interaction term, duration of hearing impairment (DHI), and duration of unaided hearing impairment (DuHI). The coefficients from the previous model [6] were fixed.

	Estimate	Standard Error	t Statistic	*p*
Constant, $\beta_{0}^{\prime }$.	0.35	0.04	8.44	3 × 10^−17^
DHI [year^−1^]	−0.0027	0.0019	−1.41	0.16
DuHI, $\beta_{4}$ [year^−1^]	−0.0171	0.0056	−3.05	0.002
DHI:DuHI [year^−2^]	−4.20	0.0001	−0.41	0.68

Included are 6600 observations, 6596 error degrees of freedom. χ^2^ statistic vs. constant model: 152, *p* = 1 × 10^−32^.

**Table 4 jcm-12-06215-t004:** Results of regression analysis with one additional predicting variable only: duration of hearing impairment (DHI). The coefficients from the previous model [6] were fixed.

	Estimate	Standard Error	t Statistic	*p*
Constant, $\beta_{0}^{\prime }$.	0.41	0.04	10.75	6 × 10^−27^
DHI, $\beta_{4}$ [year^−1^]	−0.0125	0.0013	−9.73	2 × 10^−22^

Included are 6600 observations, 6598 error degrees of freedom. χ^2^-statistic vs. constant model: 96.4, *p* = 1 × 10^−22^.

## Data Availability

Research data are available on request from the first author.

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
