# Peer review of "Evolving a Model for Cochlear Implant Outcome"

_jcm, 2023, doi:10.3390/jcm12196215_

Round 1
Reviewer 1 Report
The MS describes the extension of a previously published model of CI outcomes based on pre-operative measures. This work is timely and important because the ability to provide appropriate guidance to prospective CI patients is an ongoing issue, particularly with the large variability in outcomes that is observed. However, it is less clear how much of an improvement in the previous model this work really represents.
Major:
It is unclear whether including both groups in a single GLM is really the best strategy, or whether better predictions would be achieved by doing separate GLMs on the two groups.
It is not worthy that Group 2 had a relatively large number of WRSS65CI= 0 (Fig 4). This quite striking difference does not appear to have been discussed, nor potential reasons explored.
Author Response
Dear Editor, dear reviewers, thank you for your valuable comments on our manuscript. We revised the paper according to your suggestions and we hope that we have covered all the issues raised sufficiently. Below we explained in detail the changes. Within the manuscript we marked the changes in colour.
Best regards on behalf of the authors
Ulrich Hoppe
REV1:
Comment: The MS describes the extension of a previously published model of CI outcomes based on pre-operative measures. This work is timely and important because the ability to provide appropriate guidance to prospective CI patients is an ongoing issue, particularly with the large variability in outcomes that is observed. However, it is less clear how much of an improvement in the previous model this work really represents.
Major:
Comment: It is unclear whether including both groups in a single GLM is really the best strategy, or whether better predictions would be achieved by doing separate GLMs on the two groups.
Reply: Thank you for this comment. From the viewpoint of modelling, a model for specific subgroups is the best. However, when thinking about clinical application, a model should cover as much patient characteristics as possible. The previous dichotomous separation in groups with WRSmax>0 and WRSmax=0 was a first step and rather rough. Additionally, the limited number of patients in these subgroups limits the value of the model. Therefore, our intention was to demonstrate that a single model is able to predict postoperative speech perception at least at some degree. We now introduced a short discussion of this point in the revised manuscript.
Comment: It is noteworthy that Group 2 had a relatively large number of WRS65(CI)= 0 (Fig 4). This quite striking difference does not appear to have been discussed, nor potential reasons explored.
Reply: Yes, indeed this large number is noteworthy. Subjects belonging to group 2 have nor preoperatively measurable speech recognition up to speech levels of 120 dB. Hence, from audiometry no information about the functional integrity of the auditory nerve is known and worse results than for group 1 are to be expected. For the total group 5% (n=7) of the subjects resulted in WRSCI=0%. These seven cases have had a duration of unaided hearing loss of in average 25 years and a large degree of asymmetry. For example, three subjects suffered from single sided deafness. This may be the cause for rejecting hearing aids. Since we don’t want to be speculative we did not discuss this issue in the manuscript. These outliers will be subject of future studies. This was now added in the revised manuscript (Discussion, second paragraph).
Reviewer 2 Report
I found your work interesting and important for the prediction of CI's outcome. Please see my comments below:
1) your abstract must be 200 words only, it's not
2) You don't provide enough info about the limitations of your study since you don't include DoD and talk only about patients with <80 dB HL. It is better to create a new paragraph focusing on this topic
3) your manuscript needs proofreading there are many mistakes around
4) What was your rehab programme for 6 months? (provide more details)
5) How did you exclude patients with possible cognitive disorders since the ages of your participants (66 ± 14 years ) are among the candidates for CD or mild cognitive disorders?
6) How did you exclude the possibility of the DuHI patients during the study having any cognitive impairment?
7) Since DoD is crucial regardless of the degree of HL there is no reason to exclude subjects with a hearing threshold better than 80 dB HL. As you have mentioned, "42% exhibited a 4FPTA of ≥95 dBHL in the ear to 321 receive the implant ", is this 42% small?
needs proofreading since there are many mistakes in the manuscript
Author Response
Dear Editor, dear reviewers, thank you for your valuable comments on our manuscript. We revised the paper according to your suggestions and we hope that we have covered all the issues raised sufficiently. Below we explained in detail the changes. Within the manuscript we marked the changes in colour.
Best regards on behalf of the authors
Ulrich Hoppe
REV2:
Comment: your abstract must be 200 words only, it's not
Reply: We count 195 words and think that we are in the allowed range.
Comment: You don't provide enough info about the limitations of your study since you don't include DoD and talk only about patients with <80 dB HL. It is better to create a new paragraph focusing on this topic.
Reply: This may be a misunderstanding: We extended the model for all degrees of puretone hearing loss. We added now a remark on this in the introduction.
Comment: your manuscript needs proofreading there are many mistakes around
Reply: We revised the manuscript carefully and engaged a native speaker worked before first submission. We now checked again for remaining errors.
Comment: What was your rehab programme for 6 months? (provide more details)
Reply: The standard rehabilitation lasts for one to two years. At six months the rehabilitation is not at the end. On the one side, most of the patients reach a stable level at six months. On the other side, some of the patients needs more time and it is possible to intensify or modify rehabilitation at this time for the next months. This is now described in the revised manuscript (methods).
Comment: How did you exclude patients with possible cognitive disorders since the ages of your participants (66 ± 14 years ) are among the candidates for CD or mild cognitive disorders?
Reply: We did not screened for mild cognitive disorders explicitly. However, the clinical assessment of cognitive status is part of the CI assessment. This ensures that clinical relevant cognitive disorders are excluded. Additionally, we focused on speech recognition in quiet. Recent studies showed that minor cognitive impairment has no influence on speech in quiet [Kronlachner et al. 2018].
Comment: How did you exclude the possibility of the DuHI patients during the study having any cognitive impairment?
Reply: See above
Comment: Since DoD is crucial regardless of the degree of HL there is no reason to exclude subjects with a hearing threshold better than 80 dB HL. As you have mentioned, "42% exhibited a 4FPTA of ≥95 dBHL in the ear to receive the implant ", is this 42% small?
Reply: At first, we did not exclude subjects with hearing thresholds better than 80 dB HL. As shown in figure 3 about one third has hearing thresholds better than 80 dB. Secondly, 42% is neither big nor small. We included this number in order to enable a reference to the most recent WHO classification scheme. We revised the corresponding paragraph.